# Hydrodynamic spin-orbit coupling in asynchronous optically driven micro-rotors

Alvin Modin [1,2,4], Matan Yah Ben Zion [1,3,4] ✉ & Paul M. Chaikin[1]

Vortical flows of rotating particles describe interactions ranging from molecular machines to atmospheric dynamics. Yet to date, direct observation of the hydrodynamic coupling between artificial micro-rotors has been restricted by the details of the chosen drive, either through synchronization (using external magnetic fields) or confinement (using optical tweezers). Here we present a new active system that illuminates the interplay of rotation and translation in free rotors. We develop a non-tweezing circularly polarized beam that simultaneously rotates hundreds of silica-coated birefringent colloids. The particles rotate asynchronously in the optical torque field while freely diffusing in the plane. We observe that neighboring particles orbit each other with an angular velocity that depends on their spins. We derive an analytical model in the Stokes limit for pairs of spheres that quantitatively explains the observed dynamics. We then find that the geometrical nature of the low Reynolds fluid flow results in a universal hydrodynamic spin-orbit coupling. Our findings are of significance for the understanding and development of far-from-equilibrium materials.

Hydrodynamically coupled rotors describe the dynamics of physically diverse systems—the kinetics of proteins in biological membranes[1], the interactions of topological defects in superfluid Helium[2], and the mating rituals in dancing algae[3]. An isotropic fluid made of rotating particles with broken parity and time reversal symmetry is expected to possess peculiar material properties, including odd viscosity[4], and topological acoustic edge modes[5,6]. Moreover, simulations show that in two-dimensional systems, hydrodynamically coupled rotors self-assemble into both random and ordered hyper-uniform arrangements[7–9].

Recently, mass-produced magnetic micro-particles rotated by an external electromagnet have been extensively used as a model system for studying rotating ensembles. The rotating external field directly spins the particles, revealing new collective dynamics, including dislocation kinetics in rotating crystals[10], propagation of chiral surface waves in a rotating liquid[11], as well as self-healing and coarsening of colloidal fluids and crystals[12]. However, magnetic rotors are not free. Just like compass needles, the orientations of the magnetic dipoles are enslaved to the globally imposed north, as they are synchronized with the orientation of the electromagnet, $\theta_M$. Though the position of the center of particle $i$, $\mathbf{R_i}$, may freely diffuse in the plane, the ensemble is not isotropic, as the orientational degree of freedom, $\theta_i$, is externally imposed such that $\langle\theta_i(t)\rangle\approx\theta_M(t)$.

An alternative to magnetic rotors is particles spun by a focused beam of circularly polarized light. Photonic angular momentum can be transferred to a micro-particle through its shape anisotropy[13], birefringence[14–16], or simply by absorption[17]. Unlike magnetic rotors, photonic rotors do not directly follow the rapidly rotating electromagnetic field. Instead, the optical angular momentum flux creates a torque that maintains a steady rotation, asynchronous from the external drive. However, when using a focused beam, the position of an optical rotor, $\mathbf{R}$, is limited by the strong tweezing force at the focal point, $\mathbf{R_L}$, constraining its translational degrees of freedom $\mathbf{R}(t)\approx\mathbf{R_L}$[18–20]. For truly free rotors, both the orientations and the positions are free dynamic variables, and when in a liquid, $\mathbf{R}$ and $\theta$ are expected to couple hydrodynamically[21–23]. The motion of synthetic

[1]Center for Soft Matter Research, Department of Physics, New York University, 726 Broadway Avenue, New York, NY 10003, USA. [2]Department of Physics and Astronomy, Johns Hopkins University, Baltimore, MD 21218, USA. [3]School of Physics and Astronomy, and the Center for Physics and Chemistry of Living Systems, Tel Aviv University, Tel Aviv 6997801, Israel. [4]These authors contributed equally: Alvin Modin, Matan Yah Ben Zion. ✉ e-mail: matanbz@gmail.com

micro-rotors studied so far is incompatible with the $\mathbf{R}-\theta$ hydro-dynamic coupling that was analytically explained[3], and empirically observed[3,24] in pairs of biological micro-rotors. Moreover, previous studies with ensembles of synthetic micro-rotors[10,11] spin by an externally imposed field and can not show spontaneous symmetry breaking as seen in ensembles of biological rotors[24]. To date, configuration space is reduced in either magnetic or photonic rotors, and the nature of the chosen drive obscures mutual hydrodynamic coupling.

In this communication, we show that optically driven rotors in a non-tweezing beam freely diffuse while spinning asynchronously. By developing a novel experimental test bed that drives hundreds of free micro-rotors (Fig. 1a, b and Supplementary Movie 1), we measure their stochastic translational and rotational dynamics independently. We find that in this system, remote particles are rotating asynchronously, and at close proximity, rotation and translation couple and rotor pairs mutually advect into an orbital motion (Supplementary Movie 2). We observe that the translation and rotation of these free optical rotors reciprocate—their spin coupling obeys a geometrical relation following the Stokes flow of spheres near a wall. To create a system of asynchronous rotors, we design an optical setup capable of producing a uniform torque field with minimal tweezing. We also develop a synthetic route for stable silica-coated birefringent particles (Fig. 2) that rotate in a circularly polarized collimated beam. We characterize the translational and rotational motion of individual particles and pairs of particles and derive an analytical hydrodynamic model that quantitatively captures their dynamics.

## Results and discussion
### Synthesis of stable birefringent micro-particles
We couple photonic angular momentum to the particles by synthesizing a new type of birefringent colloid made of silica-coated vaterite.

Vaterite has a hexagonal symmetry with a positive uni-axial optical response and birefringence of $\Delta n = n_e - n_o = 0.1$, where $n_o = 1.55$ and $n_e = 1.65$ are the refractive indices along the ordinary and extraordinary axes. When illuminated with circularly polarized light, vaterite particles begin to rotate while experiencing negligible thermal absorption[16], making high particle concentrations experimentally accessible without overheating (Fig. 1b). Using previous synthetic routes[25,26], we found colloidal vaterite's rotational dynamics to be inconsistent between tweezed and tweezing-free optical fields. In the absence of tweezing, particle rotation was intermittent and gradually diminished, suggesting that minor surface chemistry variations dominated the dynamics. We develop an alternative synthetic strategy to allow consistent rotation over prolonged durations.

Particles are synthesized via controlled precipitation of highly concentrated solutions of calcium chloride $CaCl_2$ and sodium carbonate $Na_2CO_3$ according to,

$$CaCl_{2(aq)} + Na_2CO_{3(aq)} \rightarrow CaCO_{3(s)} + 2NaCl_{(aq)}. \tag{1}$$

The vaterite phase, a product of Eq. (1), is a metastable polymorph of calcium carbonate. Deviations towards even weakly acidic conditions cause rapid dissolution and transformation of vaterite spheres to calcite cubes[27]. Therefore, the vaterite-to-calcite phase transition is a significant barrier encountered when synthesizing and re-suspending vaterite microspheres.

To preserve micro-particles in the vaterite phase, we control the pH of their solution and repeatedly coat particles with silica (Fig. 2). Synthesis solutions are buffered to pH = 9.5 by n-cyclohexyl-2-aminoethanesulfonic (CHES) acid (see "Methods" for details). A typical yield for these conditions results in a cloudy suspension of billions of particles with a mean size of $3.5 \pm 0.8$ μm (Supplementary Fig. 3). We

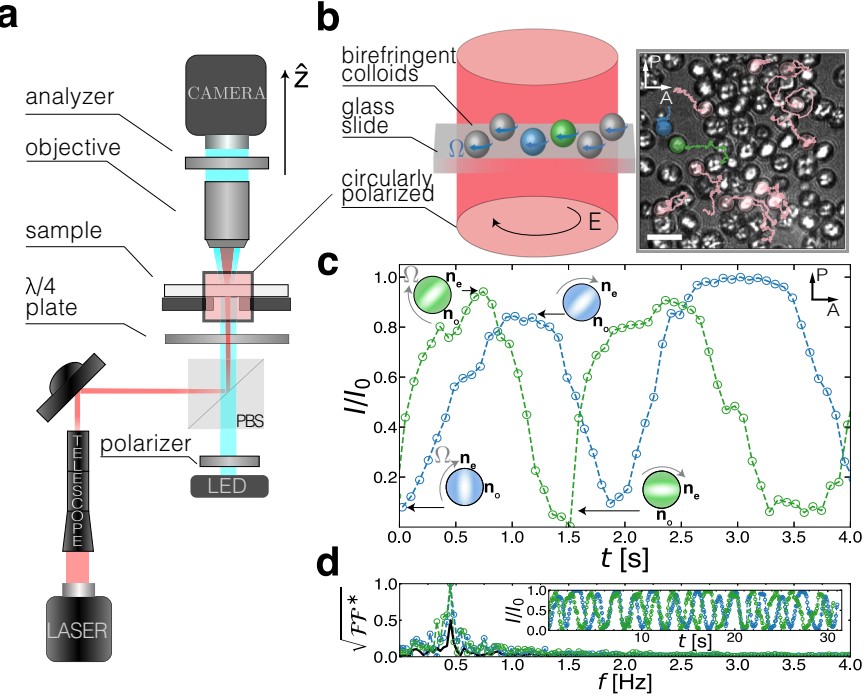

**Fig. 1 | Experimental setup to drive micro-rotors asynchronously. a** Optical setup for introducing a broad ($D \approx 440$ μm) circularly polarized beam into a microscope sample. **b** Schematic and polarized microscopy image of birefringent vaterite particles rotating while moving freely in the illuminated region. The transmitted light intensities of two particles (blue and green) are tracked over the duration of the experiment and are shown in (**c**) and the inset of (**d**). **c** One-half of the particles' (blue and green) blinking cycle, demonstrating that their optical axes are asynchronous. The incident electric field—whose direction is set by the orientation of the polarizer (P)—is de-polarized whenever the optical axis of the rotating particles is aligned with neither the polarizer nor analyzer (A). **d** Computing the magnitude of the Fourier transform ($\sqrt{\mathcal{F}\mathcal{F}^*}$) of the blinking patterns (inset) of the two particles in (**b**) shows that the frequencies at which the particles de-polarize the incident L.E.D. light are centered around 0.5 Hz, corresponding to a rotation frequency of 0.125 Hz. The magnitude of the sum of transforms, $|\sum_i \mathcal{F}_i|^2$ (solid line), decays, confirming that the particles' orientations are out of phase. Scale bar: 5 μm.

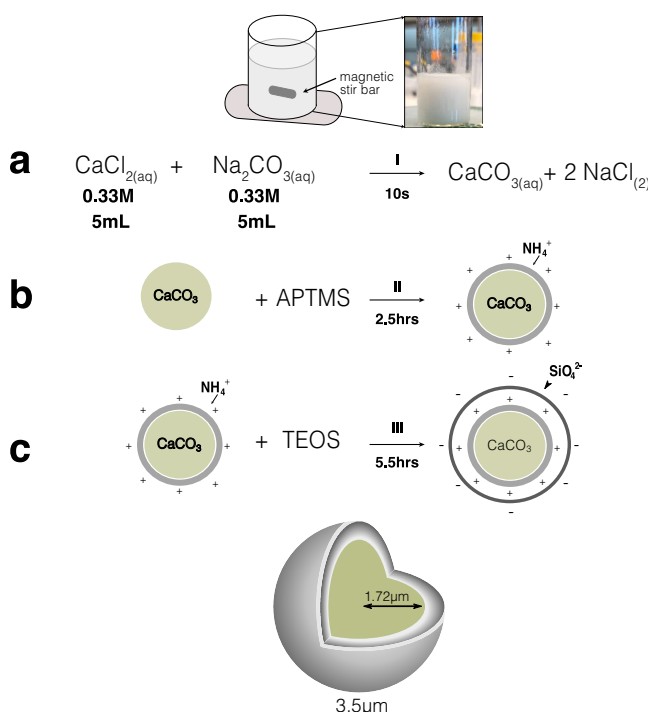

**Fig. 2 | Synthesis of silica-coated vaterite particles.** Synthesis procedure starting with (**a**) mixing of equal volumes of buffered $CaCl_2$ and $Na_2CO_3$ for 10 s with a magnetic stir bar, followed by a two-step coating procedure starting with APTMS. **b** The $CaCl_2$-APTMS solution was placed in a shaker for ~2.5 h. Particles were then coated with (**c**) TEOS and placed in a shaker for 5.5 hours. The two-step coating procedure was then repeated.

control the particles' size by varying the stirring speed, initial reactant concentration, and reaction time[28]. To further promote phase stability in solution, limit flocculation, and preserve vaterite's optomechanical behavior, particles are coated with silica by the addition of (3-amino-propyl)trimethoxysilane (APTMS) followed by tetraethyl orthosilicate (TEOS, see Methods and Fig. 2b, c[26]). Repeated silica precipitation alters the particles to be inert with long-term stability at room temperature. Typical experiments are performed in heavy water ($D_2O$), selected for its lower absorption of infrared radiation.

### Translational dynamics of individual rotors

When the laser is turned off, vaterite micro-spheres sediment onto the glass surface ($\rho_{\text{vaterite}} = 2.54$ g/cc) and diffuse in a quasi-two-dimensional plane. The gravitational height is $h_g = k_B T/F_g \approx 3 - 20$ nm, (where $T$ is the absolute temperature, and $F_g = \pi g d^3 \Delta\rho/6$ is the buoyant force given by the buoyant density $\Delta\rho = \rho_{\text{vaterite}} - \rho_{D_2O}$, and $k_B$ is the Boltzmann constant), for particles within the examined size range ($d \approx 3 - 6$ μm). Measuring the particles' mean squared displacements (MSDs), $\langle \Delta r^2 \rangle = 4 D_t \tau$, we observe a reduction in their translational diffusion constants $D_t$ compared to the bulk value, $D_t^{\text{bulk}} = k_B T/3\pi\eta d$ (Fig. 3b)[29,30]. For each particle, the mean proximity to the surface, $h$, is given by the gravitational height, and in the lubrication limit ($\langle h \rangle = h_g \ll d$) reduces the translational mobility[31]. The expected translational diffusion coefficient is

$$D_t \approx \frac{5 k_B T}{8\pi\eta d \log(d/2h)}. \tag{2}$$

To create a tweezing-free optical torque field, we use a collimated, circularly polarized infrared (IR) laser ($\lambda = 1064$ nm). When incident onto the sample p, the wide-field $D \approx 440$ μm spot delivers a maximum

power flux, $J$, of up to 40 MW m$^{-2}$. The flux varies spatially by less than ±10% within the field of view (165 μm × 125 μm), assuring that our collimated beam is free of sharp intensity gradients otherwise found in focused beams (Supplementary Fig. 1)[32]. When gradients in the beam's intensity are present, particles are constrained to the narrow waist of the focused light. A narrow-waisted beam generates tweezing forces that typically restrict the translational motion of a particle to roughly its diameter, obscuring the coupling between translation and rotation. To date, focused light beams used to rotate particles were too tight to host an ensemble of particles, making hydrodynamic particle-particle interactions inaccessible[25,33].

Our setup generates a photonic torque that drives particles to rotate while freely diffusing in-plane (Fig. 1a, b). In contrast to previous systems relying on optical tweezing or trapping[15,34], the translational MSD of rotating particles remains linear in time, $\langle \Delta r^2 \rangle \propto \tau^1$ (Fig. 4a), indicating that the broad beam profile has minimal transverse tweezing. The translational diffusion constant, $D_t$, is enhanced at higher fluxes. (Fig. 4b). Radiation pressure from back-scattered photons generates a force $F_{rad}$ opposite to gravity and smaller in magnitude. The force from the radiation pressure opposes the gravitational pull, $F_g$, effectively increasing the gravitational height to $h_g = k_B T/(F_g - F_{rad})$. The increase in $h_g$ reduces the wall's effect on the drag coefficient, increasing the translational diffusion constant until it approaches its bulk value.

At higher fluxes where $F_{rad}$ exceeds $F_g$, vaterite particles begin to steadily rise from the capillary's bottom surface at a constant speed. Vertically shifting the imaging focal plane from the bottom of the capillary to its top surface, we monitor the time it takes for particles to travel 100 μm, corresponding to when a focused image of a particle re-appears (see Supplementary Note 4 for details). We measure the force from radiation pressure $\langle F_{rad} \rangle = RJ\pi d^2/4c$ by computing the particles' vertical rising speeds and extracting the intrinsic reflection coefficient from radiation pressure, $R = 0.22 \pm 0.01$ (Supplementary Fig. 2). For a power flux of 40 MW m$^{-2}$ incident on a $d \approx 3$ μm vaterite particle, the gravitational height will increase from ≈15 nm to ≈180 nm resulting in ≈90% increase in the translational diffusion, $D_t$, consistent with measured diffusion coefficient as extracted from the MSD (Fig. 4a, b).

### Rotational dynamics of individual rotors

We measure the stochastic rotational diffusion with no drive by monitoring the transmitted light intensity of each particle. While confined to a two-dimensional plane, vaterite particles undergo rotational diffusion. When imaged under crossed-polarizers (PA), the particles' birefringence modulates the intensity of the scattered light. The fluctuations of the transmitted light intensity lead to temporal decorrelation, $g_{PA}(\tau)$, that holds information about the particle's orientational diffusion. The decorrelation rate depends on the rotational diffusion matrix $\mathbf{D}_r$, which near a wall is dominated by the spinning diffusion (rotation axis is perpendicular to the wall), $g_{PA} = \exp(-6\mathbf{D}_r\tau) \approx \exp(-6D_{r,\perp}\tau)$ (see Fig. 3c, inset and the Supplementary Note 2)[35,36]. To leading order, the diffusive spinning approaches its bulk value

$$D_{r,\perp} \approx \frac{k_B T}{\pi\eta d^3}, \tag{3}$$

and is consistent with the experimentally measured diffusion constants in the particle size range studied (Fig. 3c). The lubrication flows responsible for the reduction in translational diffusion (2) have little effect on the particle's spinning relative to their bulk dynamics (3). This relation is significant for the spin-orbit coupling of a pair of rotors.

When illuminated with circularly polarized light, individual particles rotate at a steady angular speed of up to $d\theta/dt \equiv \Omega \approx 1.2$ rad s$^{-1}$. Diffusive spinning is dominated by the external drive, with the average rotational Péclet, Pe$^r = \Omega/D_r \approx 25$. For a given particle, the spinning

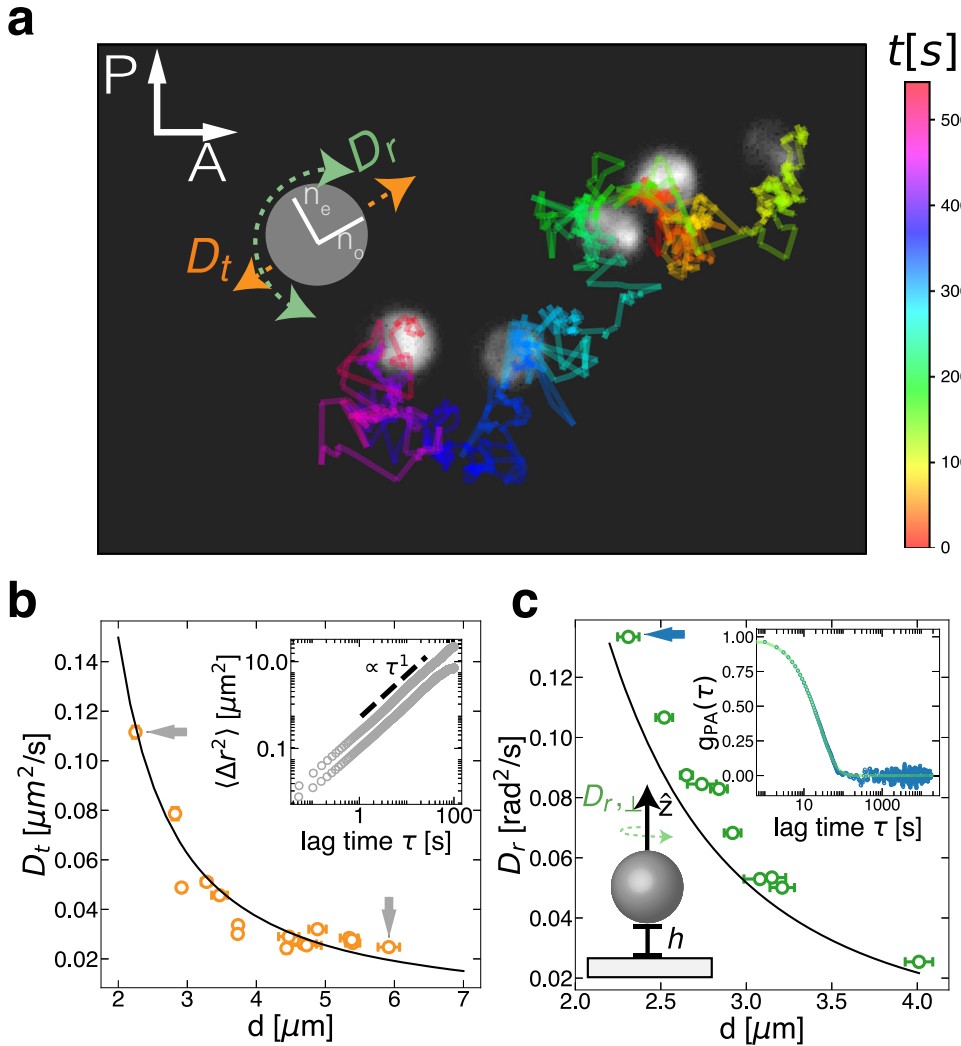

**Fig. 3 | The translational and rotational diffusion of an individual particle is described by a sphere near a non-slip wall in Stokes flow. a** Snapshots (and time-colored trajectory) of a birefringent vaterite particle viewed through crossed-polarizers, showing Brownian translation and rotation. **b** Near a no-slip wall (solid line), particles have lower translational diffusion $D_t$ relative to particles in bulk, consistent across a measured size range of $d = 2 - 6\,\mu m$. Inset shows the translational mean-squared displacement (MSDs) for particles with sizes $d = 1.92 \pm 0.11\,\mu m$

and $5.92 \pm 0.35\,\mu m$. The translational diffusion constants $D_t$ obtained from these two MSDs are indicated by gray arrows in the main panel. **c** Rotational diffusion perpendicular to the wall, (spinning) $D_{r,\perp}$, measured using depolarization intensity decorrelation, $g_{PA}$. $D_{r,\perp}$ is largely unchanged by the presence of a no-slip wall (solid line). The blue arrow indicates $D_{r,\perp}$ obtained from fitting $g_{PA}$ for a $d = 2.31 \pm 0.14\,\mu m$ particle (inset). Scale bar: 5 μm. Error bars correspond to the standard deviation.

angular frequency is given by the balance of optical torque and viscous drag

$$\Omega = \frac{TJ\lambda}{8\pi c\eta d}\left[1 - \cos\left(\frac{2\pi\Delta n d}{\lambda}\right)\right] \quad (4)$$

where $\eta = 1.25$ mPa·s is the surrounding fluid's viscosity[37], $c$ is the speed of light in vacuum, and T is the transmission coefficient[38,39] (see Supplementary Note 5 for a detailed derivation). In the chosen wavelength, vaterite has negligible absorption[16], and in the short wavelength approximation, the transmission of the refracted rays is given by $T \approx 1 - R$. We measure the birefringence of the particles, $\Delta n$, by tuning the ellipticity of the incident beam. In the presence of polarized light with ellipticity $\phi$, the optical torque is composed of both an aligning and spinning torque. For circularly polarized light, the alignment torque vanishes. Conversely, the spinning torque vanishes for linearly polarized light. The effective birefringence of the polycrystalline vaterite colloids is then measured by considering the minimum ellipticity required to generate a net torque that overcomes the viscous torque $\tau_v$ for a given particle size (Fig. 4c). We measure the

effective birefringence as $\Delta n = 0.075 \pm 0.015$, consistent with $\Delta n$ values for polycrystalline vaterite micro-particles reported in the literature between $0.06 - 0.09$[25,26,40]. Using the measured birefringence, $\Delta n$, and transmission, T, we quantitatively predict the rotation rate of individual particles as a function of flux $J$ and size $d$ (Fig. 4d). Note that the dependence of $\Omega$ on $d$ is non-monotonic due to the effectiveness of a particle as a wave plate, peaking at the thickness of an ideal half-wave plate $d_{max} \approx \frac{1}{2}\frac{\lambda}{\Delta n} \approx 5\,\mu m$ (see Eq. (4), and Fig. 4d).

**Asynchronous rotation**

We find that particles spin asynchronously by using microscopic imaging that monitors the intensity of light transmitted through individual vaterite microspheres. Particles are imaged between crossed-polarizers using a custom-built microscope with bright field ($\lambda = 505$ nm) illumination (see Fig. 1a and Supplementary Note 1 for details). When spinning, vaterite de-polarizes the transmitted light from the LED source periodically. This occurs whenever the optical axis of the particle coincides with neither polarizer nor analyzer axes, corresponding to four depolarization peaks, or blinks, per period. (Fig. 1c). The relative orientation of an ensemble of vaterite particles is

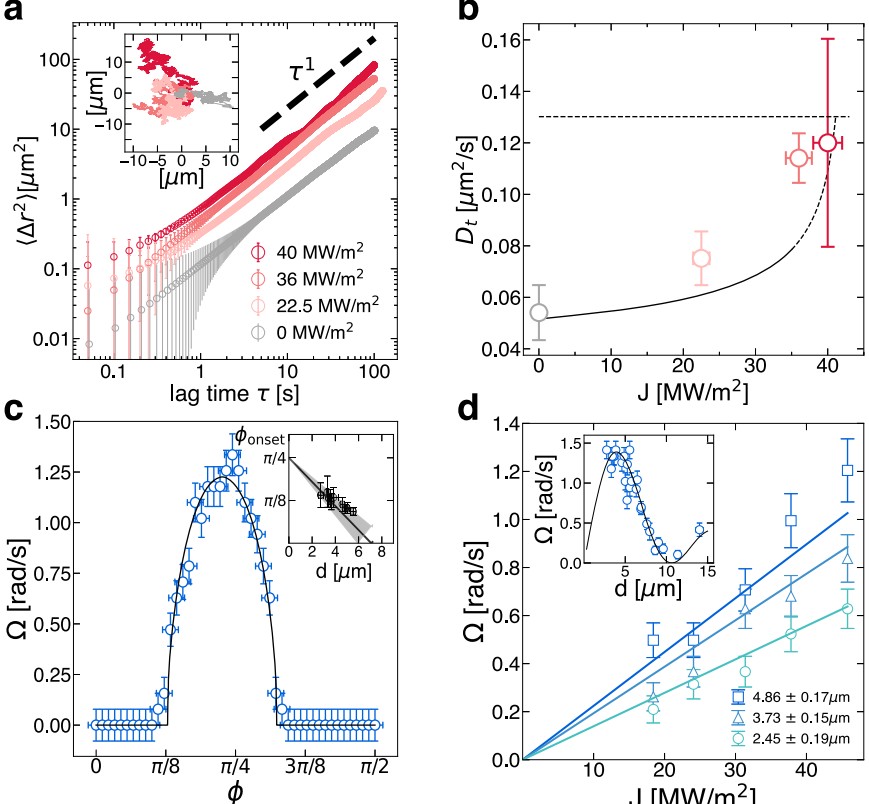

**Fig. 4 | Measurement of rotational and translational dynamics in a uniform optical torque field.** Individual rotors freely diffuse and spin in a force-free optical torque field. **a** The ensemble-averaged mean square displacement $\langle \Delta r^2 \rangle$ is linear (diffusive) for different photonic fluxes $J$. A representative trajectory of a single particle for different photonic fluxes is shown in the inset. **b** Translational diffusion increases with $J$ as the gravitational height $h_g$ increases (solid line) until it approaches the bulk value (dashed line). Colors of points in (**b**) correspond to the legend in (**a**). **c** The effective birefringence of a particle, $\Delta n$, can be measured by monitoring the minimal ellipticity, $\phi_{onset}$, where rotation begins for different particle sizes (inset). **d** Measured spin rates for different fluxes and different particle sizes (inset) as predicted by Eq. (4) (solid lines), are consistent with measured birefringence. Error bars correspond to the standard deviation.

free to vary. Tracking the transmitted light intensities of individual particles as a function of time, $I_i(t)$, allows for a direct measure of each particle's rotation frequency and phase. Simultaneously monitoring two spinning particles shows that the periodically oscillating intensities of the light they transmit are close in frequency but differ in phase (Fig. 1c). Computing the Fourier transform of the light intensities of individual rotors, $\mathcal{F}_i\big[I_i(t)\big](\omega) \equiv \int dt e^{-i\omega t} I_i(t)$, allows us to globally compare the phases of multiple particles (see Supplementary Note 6 for details). For individual particles, the magnitude of the Fourier transform, $\sqrt{\mathcal{F}_i\mathcal{F}_i^*}$ peak at $\approx 0.5$ Hz, corresponding to four times the typical particle spinning frequency ($\approx 0.125$ Hz). However, the sum of the individual Fourier transforms, $|\sum_i \mathcal{F}_i|^2$, decays with the number of particles. The different phases of the light intensities do not necessarily add up constructively, indicating that particles are globally asynchronous with respect to one another (Fig. 1d). Yet when two rotating particles approach each other, they mutually advect through their flow fields. Moreover, we observe a change in their blinking rate, indicating a change in their angular speed.

## Single particle flow field

To understand the coupling of rotor pairs, we first consider the flow field generated by a single rotor. In a uniform optical torque field, an isolated spinning vaterite micro-sphere stirs the surrounding fluid, generating an algebraically decaying flow. A single rotor can be modeled as an isolated sphere with radius $a$ centered at $\boldsymbol{r}$, subjected to a constant torque $\boldsymbol{\tau} = 8\pi\eta a^3 \boldsymbol{\Omega}^0$, where $\boldsymbol{\Omega}^0$ is the angular velocity of the isolated rotor (Fig. 5). A multipole expansion well approximates the resulting flow generated by the sphere's rotation. We introduce a

singularity at position $(x_0, y_0, z_0)$, acting as a point-torque disturbance (rotlet). The corresponding Green's function, $\mathbf{G}_{ij}$, satisfying the Stoke's equations is $\mathbf{G}_{ij} = \frac{\epsilon_{ijk} r_k}{r^3}$[41], where $\epsilon_{ijk}$ is the Levi-Cevita symbol whose indices represent components of the rotlet's position in the Cartesian coordinate system, and $r$ is a 3D vector pointing from the rotlet to a point $(x, y, z)$ in space. In an unbounded 3D Stokes fluid, the magnitude of the far-field flow of a force monopole (a Stokeslet) is $|\mathbf{u}_{Stokelet}^{bulk}| \propto 1/r$, and a force dipole $|\mathbf{u}_{Dipole}^{bulk}| \propto 1/r^2$. To leading order, the rotors are force-free but experience a torque. The resulting flow is given by an anti-symmetric derivative of the Stokeslet (a rotlet) that decays as $|\mathbf{u}_{rotlet}^{bulk}| \propto 1/r^2$.

In all experiments, the micro-rotors are found near a solid wall, imposing a no-slip boundary condition ($\mathbf{u}(z=0) = 0$) Fig. 5. Their flow field obtained by the method of images decays like the following derivative ~ $1/r^3$[41]. Explicitly, we may approximate the far-field fluid flow in the presence of a no-slip boundary as the superposition of two rotlets with equal and opposite torques $\boldsymbol{\tau} = \pm \boldsymbol{\tau}^0$ located at $z = \pm \delta$, respectively (Fig. 5). The resulting fluid flow is then $\mathbf{u}(r) = \Omega^0 a^3 \left\{ \left( \frac{1}{R_+^3} - \frac{1}{R_-^3} \right)(-y\hat{\mathbf{x}} + x\hat{\mathbf{y}}) \right\}$ (see Supplementary Note 7 for a detailed calculation). Here $\hat{\mathbf{x}}, \hat{\mathbf{y}}$ are Cartesian unit vectors, and $|R_\pm| \equiv \left( x^2 + y^2 + (z \mp \delta)^2 \right)^{\frac{1}{2}}$, representing the distance to a point $(x, y, z)$ in space from the source and image charges, respectively. In the far-field limit, $|R_\pm|^{-3} \approx \frac{1}{r^3}\left(1 \pm \frac{3\delta^2}{r^2}\right)$. When close to the wall ($\delta \approx a$), the $1/r^3$ contribution vanishes, and the next term in the multipole expansion is now proportional to $1/r^4$. This scaling arises from noting that

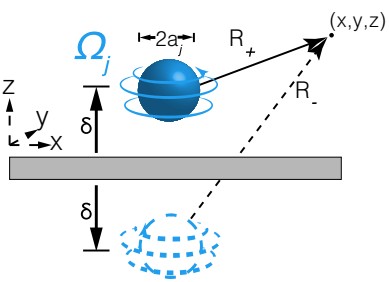

**Fig. 5 | Flow field of a single rotor spinning near a wall.** Geometry of a rotating sphere and its corresponding image charge generated near a solid no-slip wall while rotating about an axis perpendicular to the wall.

$-y\hat{\mathbf{x}} + x\hat{\mathbf{y}} = r\hat{\boldsymbol{\theta}}$. To leading order, a sphere spinning at an angular frequency $\Omega$ near a wall generates a flow of,

$$\mathbf{u}(\mathbf{r}) = \frac{6\Omega^0 a^3 \delta^2}{r^4}\hat{\boldsymbol{\theta}}, \tag{5}$$

where $r$ is now the distance in the two-dimensional plane. The reciprocal motion of a pair of rotors can be described as though each generates the flow field in Eq. (5) while being advected by the same flow profile generated by the companion rotor.

### Hydrodynamic spin-orbit coupling in a pair of micro-rotors

When two micro-particles are sufficiently close to each other, their long-range flow fields drive them into a short-lived orbit until they diffuse apart. In a typical interaction, a pair of particles with a mean size ratio 0.75–1, exhibit relative orbital motions for 20–30 s before separating (Fig. 6 and Supplementary Movie 2). As two rotors orbit each other, each particle's angular speed, $\Omega$, is reduced from its nominal value (Fig. 6c, inset). When in close proximity, the blinking rate of the particles can decrease by up to 35%, corresponding to a slowdown in their spinning rate. Yet particles continue to rotate asynchronously even at near contact (Supplementary Movie 2). Being of finite size, a particle's rotation rate changes when subjected to flow gradients. The orbital interaction of the synthetic photonic rotors resembles the dynamics seen in microscopic organisms[3,24,42], and stands in stark contrast to interactions seen in magnetically rotated particles[12].

The dynamics of this spin-orbit interaction can be described with a minimal far-field hydrodynamic model (see Supplementary Note 7 for a detailed calculation). For pairs of particles $i$ and $j$ (Fig. 6a), the velocity of a spherical particle $i$, $\mathbf{v}_i$ found within the flow field of particle $j$, $\mathbf{u}_j$ is given by Faxen's first law[43,44]

$$\mathbf{F}_i = 6\pi\eta a\left\{\left[\mathbf{u}_j(\mathbf{r}) + \frac{1}{6}a^2\nabla^2\mathbf{u}_j(\mathbf{r})\right]_{\mathbf{r}=\mathbf{r}_i} - \mathbf{v}_i\right\}, \tag{6}$$

where $\mathbf{F}_i$ is the external force. In our case, particles are force-free, $\mathbf{F}_i = 0$, and $\nabla^2\mathbf{u}$ is vanishing, so to leading order particle $i$ is simply advected by particle $j$: $\mathbf{v}_i \approx \mathbf{u}_j(\mathbf{r}=\mathbf{r}_i)$. The measured orbital frequency $\boldsymbol{\omega}_i \equiv \mathbf{v}_i/r$ as a function of particle separation $\mathbf{r}$ for a pair of interacting particles is then quantitatively captured by combining Eqs. (5) and (6) (Fig. 6c). It is further known that the flow-induced angular velocity is proportional to the vorticity. For particle $i$, the expected change in the rotation rate, $\Delta\Omega_i$, caused by the flow generated by particle $j$ is given by Faxen's second law[43], connecting the apparent spinning of particle $i$, $\boldsymbol{\Omega}_i$ with the flow generated by particle $j$, $\mathbf{u}_j$

$$\boldsymbol{\Delta\Omega}_i \equiv \boldsymbol{\Omega}_i - \boldsymbol{\Omega}_i^0 = \frac{1}{2}\nabla\times\mathbf{u}_j(\mathbf{r}=\mathbf{r}_i). \tag{7}$$

For identical rotors, $\boldsymbol{\omega} = 2\boldsymbol{\omega}_i$ is the orbital frequency of the rotating pair about their common center. Recalling that rotating particles generate

an algebraically decaying tangential flow field ($\mathbf{u} \propto 1/r^\alpha\hat{\boldsymbol{\theta}}$), Eq. (7) becomes

$$\boldsymbol{\Delta\Omega} = \frac{1}{4}(1-\alpha)\boldsymbol{\omega}, \tag{8}$$

connecting the spin angular frequency change to the rotating pair's orbital frequency. The above derivation follows closely the hydrodynamic description of biological micro-rotors near a no-slip boundary[3] while keeping $\alpha$ implicit, which only changes the slope of the linear spin-orbit coupling. The relation in Eq. (8) is general—independent of $a$, $r$, and $\delta$. Every translation is accompanied by a proportional amount of rotation. Eq. (8) holds regardless of whether the flow is three-dimensional (in bulk), quasi-two-dimensional (near a wall), or strictly two-dimensional (in a liquid film). As expected from Stokes flow, spin-orbit coupling is geometrical in nature and does not depend on physical parameters, such as the applied torque, fluid viscosity, and material composition. Surprisingly, the far-field approximation quantitatively captures the orbiting dynamics even when particles are nearly in contact. In our case, $\alpha = 4$ (particles are near a wall), and as seen in Fig. 6d, this relation captures the spin-orbit dynamics of optical rotors of different sizes, spinning rates, and over a range of separations. This connection was not observed in previous systems of synthetic rotors and was impossible to extract from pairs of magnetic rotors which were shown to rotate as a solid body for any rotational frequency[12].

In this work, we introduce a new system of active spinning particles —asynchronous photonic rotors enabled by a tweezing-free optical field. We design a force-free torque field using a collimated beam of circularly polarized light and develop a synthetic route for birefringent silica-coated vaterite colloids to show for the first time the spinning of hundreds of micro-particles using photonic angular momentum. We systematically quantify the micro-rotors' optical and hydrodynamic properties and found that particles rotate asynchronously, unlike any previous synthetic micro-rotor system. The particles' asynchronous rotation indicates that their orientational degrees of freedom are dynamic variables; this is in contrast to magnetic rotors, whose orientational degrees of freedom are "frozen" by an applied magnetic field. We analyze the particles' spinning rate and found that pairs of rotating particles mutually advect one another, with their translation and rotation coupled hydrodynamically. Our analysis shows that, as the coupling is geometric, it may be applicable more generally in active systems, from living organisms − where it has already been rigorously shown to be the case both empirically and analytically[3,24] − to even in robotic systems[45,46], where translation and rotation are coupled. Our system allows for further investigation into isotropic rotating ensembles with broken time-reversal symmetry and parity, shedding light on new material properties theoretically predicted in active matter such as odd viscosity and quantum hall fluids[4,6,47]. Using non-spherical particles, free optical rotors can also be used to study the effect of morphology and steric interactions in tandem with hydrodynamic coupling[48,49]. Moreover, combining our system of optical rotors with rotors driven by an external magnetic field could enable the experimental study of ensembles of counter-rotating particles, where optical rotors rotate independently from the magnetic rotors. Experimental investigation of an ensemble of counter-rotors would elucidate recent predictions on self-assembly, phase separation, and edge modes, expanding our understanding of far-from-equilibrium states of matter[23,50–52].

## Methods

### Vaterite synthesis and sample preparation

Vaterite micro-spheres are synthesized by controlled precipitation from a super-saturated solution of 0.33M calcium chloride (CaCl₂, Sigma-Aldrich) and 0.33M sodium carbonate (Na₂CO₃, Sigma-Aldrich). We buffer CaCl₂ and Na₂CO₃ to a pH of 9.5 (CHES, Sigma-Aldrich),

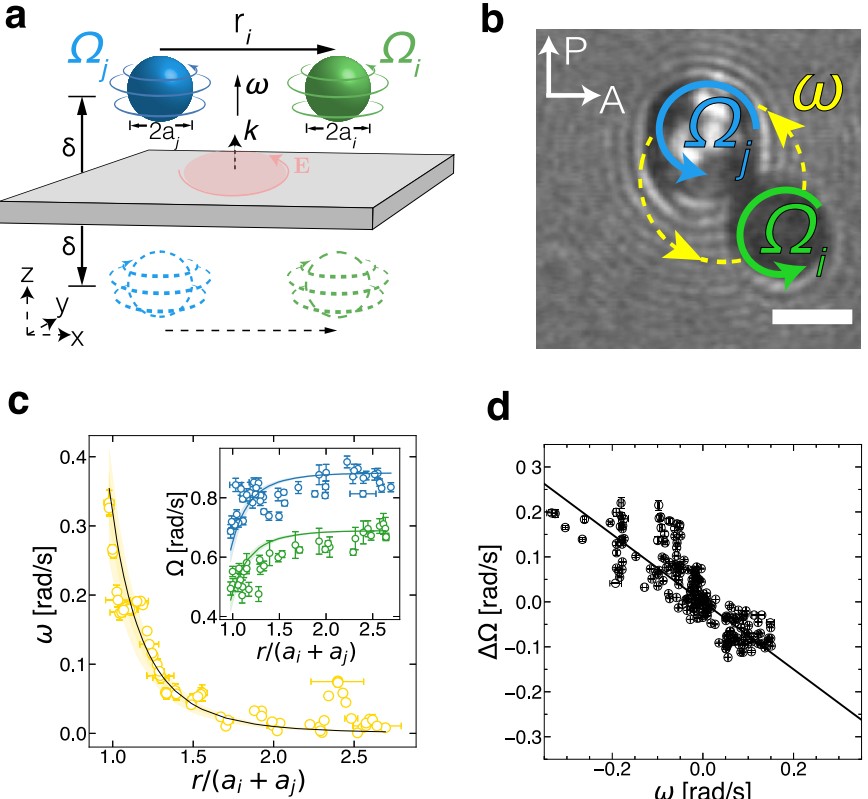

**Fig. 6 | Pairs of free rotors couple hydrodynamically through rotation and translation. a** Diagram of an orbiting pair near a wall with corresponding image charges. **b** Snapshot of two mutually advecting particles. **c** Angular speed $\omega$ of two orbiting spheres (diameters 6.7 µm and 5.1 µm) at different separations, along with the dependence of the spinning rate $\Omega$ on the normalized separation (inset). Curves show the predicted spinning rates for each particle at different separations− derived from Faxen's laws (Eqs. 6 and 7 in the main text)− given the particles'

asymptotic spinning rate, $\Omega_{i,j}^0$, measured at large separations (colors correspond to panel **b**). **d** The change in spin $\Delta\Omega$, and the orbital frequency $\omega$ for rotors of varying size, separation, and optical flux follows a geometric relation given by Eq. (8), where $\alpha = 4$ (solid line). The scatter relative to the trend line originates from thermal fluctuations in transient orbits of freely diffusing Brownian rotors (see Supplementary Note 7). Scale bar: 5 µm. Error bars correspond to the standard error.

before mixing at 1000 RPM in a glass vial with a 1 cm magnetic stir bar Fig. 2a. The total stirring time is ~40 s. For these conditions, the typical poly-dispersity is $3.6 \pm 0.8$ µm, although, by varying the synthesis conditions, a particle diameter range of $d = 2$–12 µm is readily accessible (Supplementary Fig. 3). Particles are then coated via a sequential coating process using (3-Aminopropyl)trimethoxysilane (APTMS, Sigma-Aldrich) and tetraethyl orthosilicate (TEOS, Sigma-Aldrich) Fig. 2. In a typical APTMS coating, 1.5 mL of the synthesis bath is washed in DI $H_2O$ 3 times, to which 70 µL of APTMS (Sigma-Aldrich), 25 µL of Ammonia (25% V/V in $H_2O$, Merck) and 940 µL of ethanol (200 proof) are added. The sample is then placed in a shaker for 2.5 h. For TEOS coatings, the procedure is identical (APTMS is replaced with TEOS), except that the sample is allowed to shake for 5.5 h. Electrostatic interactions are minimized by the presence of 14 mM NaCl in the solution, reducing the Debye screening length to 2.5 nm. Microscope samples are made by dispersing the particles in heavy water ($D_2O$, Sigma-Aldrich) and loading into a 100 µm tall glass channel (Vitrotubes W5010050) passivated through vapor deposition of hexamethyldisilazane (Sigma-Aldrich). Loaded capillaries are placed onto a clean microscope glass slide and sealed on their ends with UV-curable resin (Loon Outdoors UV Clear Fly Finish).

### Experimental setup

Imaging is performed on a custom-built, bright-field microscope coupled to a laser source. A commercial light emitting diode ($\lambda = 505$ nm Thorlabs) with a diffuser (ground glass N-BK7 600 grit, Thorlabs), condenser, and an iris are used to achieve Köhler

illumination. The scattered light is picked up by the microscope objective (HCX PL APO 40x NA = 0.85, Leica) and a tube lens (B&H), detected by a digital camera (DCC1545M, Imaging Source), and acquired using commercial video recording software (IC Capture, Imaging Source). A laser beam was introduced on a separate optical path (see Supplementary Fig. 1a). A $\lambda = 1064$ nm laser beam (YLR-10-1064-LP, I.P.G. Photonics) passes through a zero-order half-wave plate (WPH05M-1064 Thorlabs) and is contracted using a customized Galilean telescope to achieve a wide beam (Supplementary Fig. 1b). The laser beam is introduced into the sample using a polarizing beam splitter (PBS CM1-PBS253 Thorlabs). Its intensity at the sample is controlled by a combination of the electronic laser head controller and adjustment of the half-plate. The intensity is measured using an optical power meter (PM100D power meter, with S175C sensor, Thorlabs). In order to eliminate laser intensity before the camera, stained glasses (FGS900S, Thorlabs) are stacked after the objective.

### Data availability

The data generated in this study are available at an online repository[53] and can be accessed at the following URL: https://doi.org/10.6084/m9. figshare.22294690.

### Code availability

The custom codes used in this study are available from the corresponding author upon request.

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

## Acknowledgements
We greatly acknowledge insights and assistance from Joon Oh, Naomi Oppenheimer, Yoav Lahini, Nathaniel Spilka, Yasuo Oda, Bastián Pradenas, David Rivas, Yihao Chen, Sofia Magkiriadou, and Raymond Goldstein. This research was supported by the Department of Energy DE-SC0007991 for initiation and design by P.M.C. and by DOE SC0020976 for sample preparation and imaging by M.Y.B.Z. and A.M.

## Author contributions
A.M., M.Y.B.Z., and P.M.C. conceived the project. A.M. and M.Y.B.Z. designed and conducted the experiments, data analysis, and developed the theoretical model. All authors contributed to the writing of the manuscript.

## Competing interests
The authors declare no competing interests.
