## [Peer Review File · Nature Communications]

Hydrodynamic spin-orbit coupling in asynchronous optically driven micro-rotorsREVIEWER COMMENTS

Reviewer #1 (Remarks to the Author):

The manuscript presents an experimental realization of spinning spherical particles with axis of rotation normal to a bottom wall. The rotation is driven by a focused beam of circularly polarized light; I could not follow the details but this seems like a clever idea. The particle rotation rate is set by the balance of optical and viscous torques. Particle translations and rotations are measured and the authors conclude that hydrodynamic interactions give rise to rotor pairing and orbiting around each other. The authors claim “universal hydrodynamic spin-orbit coupling” which is “geometrical in nature” but, unless I am missing something, it is expected that HD interactions between solid particles in Stokes flow depend only on geometry (Kim and Karrila, Microhydrodynamics, 1991)

It is not clear what the significance of the fact that particles optical axes are asynchronous – wouldn't that just be set by the initial (random) conditions?

More details need to be provided about the model, at least in the supplemental material. Distinction should be made between the 3D r and the 2D r in Eq. 5. Eq. 8 takes the derivative of $1/r^\alpha$ to claim that the rotation rate decreases by a factor of $1-\alpha$. However, in Eq. 5 there is prefactor $3a^3 \Delta^2/r^5$. Is this really $O(1)$ constant? It does depend on the separation between the rotors r . Is this the universality proposed by the authors – that the dependence on size, separation, and Δ somehow compensate each other to give order 1 constant? If this is the case, this point should be clarified. However, the data in fig. 6d is pretty scattered and not very convincing

In a study that emphasizes the importance of HD interactions, it seems the authors ignore a significant body of work dedicated on the emergence of hydrodynamically bound states in systems of rollers, e.g., Martinez-Pedrero et al (Sci. Advances, 2018) Delmotte (Phys. Rev. Fluids, 2019), and spinners (confined to a 2D plane)-in addition to Ref. 24, Goto et al. (Nat. Comm, 2015), Kokot et al (PNAS, 2017). In addition to Ref. 7-9, the ordering in monolayer of rotors has been analyzed by Lushi and Vlahovska (J Non-linear Science, 2015), where orbital motion in such 2D systems of rotors is predicted (albeit in free space, no wall).

In conclusion, while the experiment is neat, the novelty beyond the new experiment is limited. While orbital motion is indeed observed for a first time experimentally, it has been theoretically predicted in systems of plane-confined rotors.

Reviewer #2 (Remarks to the Author):

See attached.

Reviewer #3 (Remarks to the Author):

The manuscript "Hydrodynamic spin-orbit coupling in asynchronous optically driven micro-rotors" presents a synthetic system of active self-rotating particles which have both rotational and translational degrees of freedom.

The manuscript presents a novel biomimetic system of active rotors. Other examples of synthetic active rotors have translational, but not rotational, degrees of freedom because their active rotations closely follow an external field. By contrast, in this manuscript, the particles are able to rotate with different speeds and different phases, much like self-spinning

living cells, which interact only hydrodynamically. The manuscript is well written and includes a concrete set of experimental results that convincingly test quantitative predictions for these kinds of active rotors. I recommend publication.

Before publication, I suggest that the authors consider the following two points:

- The inset in Fig. 6c was not clear to me. Is it the spinning rate Ω or its change $\Delta\Omega$ which is plotted? What are the theoretical predictions plotted along with the experimental data?

- The following recent work may be of interest as an analogous biological system:
Odd dynamics of living chiral crystals
Tan et al Nature 607, 287 (2022).

Review of *Hydrodynamic spin-orbit coupling in asynchronous optically driven micro-rotors* by A. Modin et al.

The manuscript reports a study of the dynamics of a sample of micro-particles immersed in water. The particles are birefringent, and some effort was made with particle synthesis to ensure they were stable in water. The particles can be optically rotated by illuminating them with circularly polarised light - and an unfocussed laser beam is chosen such that the particles are not translationally optically trapped, and so are still able to freely diffuse while rotating. 2D particle tracking is performed allowing the trajectories of the particles, and their hydrodynamic interactions to be analysed. This analysis compares well with analytical theory of low Reynolds number hydrodynamics. In particular, the interaction of pairs of particles is observed, in which the particles become transiently coupled in their translational motion.

I find the work interesting, and as far as I am aware, this is the first such analysis of a complex sample of this nature. The experiments and theoretical analysis appear to be rigorously carried out, however I do think the explanation of the theory could be clearer in places, see more details below. Perhaps I am missing something deeper, however it does not seem unexpected that a spinning particle should cause nearby particles to orbit. Therefore overall I am unsure of the wider significance of these findings. A few examples are given in the conclusion, however it is unclear where or how the theory developed in this work could be applied to give a deeper understanding of any other situation/type of sample. Therefore, I suggest this wider significance is articulated more clearly for consideration in a broad interest journal such as nature communications.

Small points:

line 106: MSD - Mean squared displacement? is not defined. I think should be for a broad audience journal like nat. comms.

line 113: The manuscript states ‘..monitoring the particles’ vertical rising speed..’ How is this out of plane motion monitored?

line 161: ‘The magnitude of the Fourier of the individual particles,..’ should instead be something like: ‘The magnitude of the Fourier Transform of the intensity transmitted by individual particles as a function of time,..’?

line 162: ‘corresponding to four times the typical rotation frequency (~ 0.125 Hz)’ It is not clear to me why this is four times rather than twice the typical rotation frequency?

line 163: ‘The asynchronous phases of the light intensities add-up destructively,’ I know what is meant here, however I find this use of the term ‘destructive’ when speaking of purely real positive functions which can’t ‘cancel each other out to zero’ a bit misleading. Consider re-phasing this sentence?

Equation 4 - might be helpful to give more detail of how this equation was constructed, maybe in the supplementary.

Figure 3b inset: I don’t follow why the data points appear to form into two parallel lines. Is this showing particles of different sizes? I think this inset needs more explanation in the caption/main text. Also, what sized particle is this data plotted for?

Line 173: The Green’s function equation - I suggest all variables (epsilon, r - is at distance in any direction, or distance perpendicular to the rotation axis?) and indices (ijk) are specifically defined here, along with some additional explanation, to help those readers unfamiliar with how the Stokelet and Rotlet descriptions of hydrodynamic interactions operate.

Line 178-187: I suggest clarifying this section. I don’t follow how the statement on line 179-180 about the flow field decaying as $\sim 1/r^3$ then ties in with the following sentences and eq5 which has a $1/r^4$ scaling. These equations are introduced quickly, without proper definition of the many terms (e.g. r , x^\wedge , y^\wedge , θ^\wedge). Also is R superscript(+) the same as R subscript(+)? I assume so but it is a little confusing seeing both. Figure 5 is helpful, and I understand these equations are well

used in many body low Reynolds number hydrodynamics (as is the method of images near a boundary), however for readers unfamiliar with this field, I think more explanation is needed.

Line 197: 'microscopic organism' > 'microscopic organisms'.

Response to Referees for Manuscript NCOMMS-22-37460

March 17, 2023

Response summary

Dear Referees,

We are writing to resubmit a revised version of our manuscript entitled “Hydrodynamic spin-orbit coupling in asynchronous optically driven micro-rotors” (NCOMMS-22-37460).

We thank the Referees for their kind words, finding our work “**clever**” (Referee 1) and “**rigorously carried out**” (Referee 2). We are also grateful to Referee 3, who found our synthetic rotors to be “**much like self-spinning living cells.**”

Please find below a point-by-point response for each Referee’s professional critique, separated by Referee. Referees’ questions are highlighted in **bold**. Changes in the research article are highlighted in **green** in the new version of the manuscript.

We appreciate all of the Referees’ comments and questions, helping to raise the manuscript’s scientific standards to match *Nature Communications*. Thank you for considering our re-submission.

Sincerely,

Matan Yah Ben Zion

Alvin Modin

Paul Chaikin

Response to Referee 1

1. The manuscript presents an experimental realization of spinning spherical particles with axis of rotation normal to a bottom wall. The rotation is driven by a focused beam of circularly polarized light; I could not follow the details but this seems like a clever idea. The particle rotation rate is set by the balance of optical and viscous torques. Particle translations and rotations are measured and the authors conclude that hydrodynamic interactions give rise to rotor pairing and orbiting around each other.

The authors claim “universal hydrodynamic spin-orbit coupling” which is “geometrical in nature” but, unless I am missing something, it is expected that HD interactions between solid particles in Stokes flow depend only on geometry (Kim and Karrila, *Microhydrodynamics*, 1991)

We agree that, in theory, Stokes flow is well known to be geometric in nature. However, experimental work studying hydrodynamic coupling between rotating particles thus far showed mixed contributions, including steric interactions^{1,2}, magnetic interactions³, or phoretic interactions⁴. We designed a new experimental approach for rotating micro-particles to decouple the different contributions. We did not use a focused beam of light (as was done before⁵⁻¹⁰). Instead, we used a collimated beam (as illustrated in Fig. 1 a,b). This distinction allows us to experimentally investigate previously inaccessible conditions for the following reasons:

- (a) Focused light beams create sharp light intensity gradients that tweeze particles to the tight focal point of the beam. This typically restricts the translational motion of the particle to roughly its size, obscuring the coupling between translation and rotation.
- (b) To date, the focused light beams used to rotate particles were too tight to host more than one particle at a time, making hydrodynamic particle-particle interactions inaccessible.

Using a broad, collimated beam of light, we were able to observe the geometric nature of Stokes flow directly, derive the universal hydrodynamic spin-orbit coupling, and experimentally support theoretical predictions found in Kim and Karrila (which we now cite).

The distinction between a focused and collimated photonic torque field is particularly important when testing the significance of pair interactions in search for theoretically predicted emergent behavior^{2,11-14}.

We added the following in the text to illustrate this:

When gradients in the beam’s intensity are present, particles are constrained to the narrow waist of the focused light. A narrow-waisted beam generates tweezing forces that typically restrict the translational motion of a particle to roughly its diameter, obscuring the coupling between translation and rotation. To date, focused light beams used to rotate particles were too tight to host an ensemble of particles, making hydrodynamic particle-particle interactions inaccessible^{9,15}.

2. It is not clear what the significance of the fact that particles optical axes are asynchronous – wouldn’t that just be set by the initial (random) conditions?

The Referee correctly points out that having rotors with different initial orientations would be sufficient to show the decay of the magnitude of the Fourier transform of the summed intensities (Fig 1d). This alone is an experimental observation previously inaccessible for synthetic microrotors – magnetic particles, for example, will always align their dipole moment with the direction of the applied external magnetic field. The applied field artificially “freezes” the orientational degrees of freedom of magnetic rotors.

Following the Referee’s comment, we identify the need to point out further sources of asynchronicity:

- (a) Different initial conditions
- (b) Difference in rotation frequency
- (c) Difference in the stochastic torque

The inset of Fig. 1d shows that the two oscillating signals change their relative phase. This can be interpreted as a difference in the stochastic torque experienced by either particle, indicating that the relative orientation of the particles changes with time. By contrast, magnetically rotated particles (whose orientational degrees of freedom are frozen) have the same orientation (and relative orientation) throughout an experiment. Asynchronicity is the crucial ingredient that allows observation of hydrodynamic spin-orbit coupling. If a particle’s orientation is constrained to the orientation of the external drive, it will not respond to the hydrodynamic torque generated by neighboring particles. For clarification, we added the following sentence in the main text:

The relative orientation of an ensemble of vaterite particles is free to vary.

We highlight this point once again in the conclusion of the manuscript:

We systematically quantified the micro-rotors’ optical and hydrodynamic properties and found that particles rotate asynchronously, unlike any previous synthetic micro-rotor system. The particles’ asyn-

chronous rotation indicates that their orientational degrees of freedom are dynamic variables; this is in contrast to magnetic rotors, whose orientational degrees of freedom are “frozen” by an applied magnetic field.

We have also added a section to the Supplementary Information:

- *Measuring the frequency and global phase of a rotating particle*

The section contains the following supporting information:

- (a) A detailed calculation of the Fourier transform of an individual rotor.
- (b) A detailed calculation of the Fourier transform of the sum of the transmitted light intensity of individual rotors.

3. **More details need to be provided about the model, at least in the supplemental material. Distinction should be made between the 3D r and the 2D r in Eq. 5. Eq. 8 takes the derivative of $1/r^\alpha$ to claim that the rotation rate decreases by a factor of $1 - \alpha$. However, in Eq. 5 there is prefactor $3a^3\delta^2/r^5$. Is this really $\mathcal{O}(1)$ constant? It does depend on the separation between the rotors r . Is this the universality proposed by the authors – that the dependence on size, separation, and δ somehow compensate each other to give order 1 constant? If this is the case, this point should be clarified.**

We thank the Referee for pointing out the need to clarify our model further. The relation (Eq. 8) is universal and is obtained by combining Eqs. 6 and 7 using the flow field given by Eq. 5.

The Referee is correct in noting that the pre-factor $3a^3\delta^2/r^5$ in Eq. 5 depends on the separation between the rotors, r . However, because of geometry, every translation is accompanied by a proportional amount of rotation. When we re-scale the data in Figure 6c according to the particles’ radii a , separations r , and distances from the wall of the capillary δ , we find that these parameters compensate for each other in a way that results in an $\mathcal{O}(1)$ constant. For example, the maximum value of advective flow $\mathbf{u}(r)$ experienced by a neighboring particle occurs when particles touch. Assuming two identically sized spheres (with radius a), the minimum distance r between their two centers is $r = 2a$, resulting in a pre-factor $3a^3\delta^2/r^5 \approx 3/32$.

This allows us to obtain a scaling law for spin-orbit coupling that depends only on the type of confining geometry (Eq. 8) being considered. For example, a rotor near a plane has an $\alpha = 4$, but a rotor confined between two planes or next to a fluid-fluid interface will have a different value of α and thus

a different “strength” of spin-orbit coupling. The spin-orbit scaling law depends only on the flow-field generated and not on the material parameters of the particles. When accounting for these parameters, the experimental results presented in Fig. 6d fall onto a line with a slope of $1 - \alpha = -3$, as predicted. To provide more clarity regarding the model, we have added a new section in the Supplementary Information, titled:

– *Flow generated by a rotating sphere near a wall in the Stokes-flow regime*

This section includes a step-by-step derivation of Eqs. 5, 8, and 9. In the text, we added key derivation points that emphasize the distinction between 2D and 3D r . For completeness, on lines 203-205, we define all variables as:

Here \hat{x}, \hat{y} are Cartesian unit vectors, and $|R_{\pm}| \equiv \left(x^2 + y^2 + (z \mp \delta)^2\right)^{\frac{1}{2}}$, representing the distance to a point (x, y, z) in space from the source and image charges, respectively. In the far-field limit, $|R_{\pm}|^{-3} \approx \frac{1}{r^3} \left(1 \pm \frac{3\delta^2}{r^2}\right)$.

And after Eq. 8, we have added the following:

...where r is now a two-dimensional distance.

We have also stated that Eq. 9 is explicitly independent of particle size, separation, and height above the rotating plane:

This relation is general – independent of a , r , and δ . Every translation is accompanied by a proportional amount of rotation. Eq. 8 holds regardless of whether the flow is three-dimensional (in bulk), quasi-two-dimensional (near a wall), or strictly two-dimensional (in a liquid film).

4. However, the data in fig. 6d is pretty scattered and not very convincing.

The scatter in the spin-orbit coupling measurement (Fig. 6d) can be estimated from the thermal fluctuations of a Brownian rotor. The time evolution of the variance of the orientation, $\langle \Delta\theta^2(t) \rangle$, of a rotor with a nominal spinning rate of Ω_0 , which is subjected to rotational diffusion with rotational diffusion constant D_r , follows an equation similar to a 1D Brownian particle subjected to an external drift: $\langle \Delta\theta^2(t) \rangle = 2D_r t + (\Omega_0 t)^2$ (see for example Doi, Oxford University Press, 2013). At short times ($t \ll \frac{2D_r}{\Omega_0^2}$), the motion is diffusion dominated, and at longer times ($t \gg \frac{2D_r}{\Omega_0^2}$), the motion is drift (or activity) dominated. This means that if the orientation is monitored over a short duration, we should expect inherent fluctuations stemming from the diffusive term, with their relative significance depending on the ratio of the drift to the diffusive contributions. Note that this is analogous to the Péclet

number, which is typically related to translational motion in the literature. In our work, we measured the different parameters and can estimate quantitatively that for a typical 4 μm particle, rotational diffusion is $D_r \approx 0.02 \text{ rad}^2/\text{s}$ (see Fig. 3c in the main text), and a nominal spinning rate is $\Omega_0 \approx 1 \text{ rad/s}$. Since the particles also undergo translational diffusion, their orbits are transient, limiting the duration over which the instantaneous rotation rate can be extracted. This requires striking a balance between the following extremes: on one end, if a particle’s orientation is monitored over a long duration, it will average over different orbital separations. On the other hand, if the particle’s orientation is measured over too short of a period, its dynamics will be dominated by thermal diffusion. To balance these, we extract the instantaneous spinning rate in the period between two blinks, $\tau_{\text{blink}} \approx 2\text{s}$ (see Fig. 1c). This gives a relative error in the measured “instantaneous” spinning rate of $\sqrt{2D_r\tau_{\text{blink}}/(\Omega_0\tau_{\text{blink}})^2} \approx 0.14$. By comparison, the relative contribution of the spin-orbit coupling to the instantaneous spinning rate is about $\Delta\Omega/\Omega_0 \lesssim 0.3$, as measured in our work (see Fig. 6c inset) and also theoretically predicted in the past (Davis1969). Thus, the contribution of the spin-orbit coupling to the relative change in spinning rate is larger but comparable to the relative thermal fluctuations, consistent with the scatter observed in Fig. 6d.

We thank the reviewer for raising this important point, as it emphasizes the unique dynamics of *free* rotors that undergo rotational and translation diffusion. In the revised manuscript, we added two subsections in the Supporting Information detailing the instantaneous spinning rate measurement process and evaluating their fluctuations. These subsections are titled:

- *Expected fluctuations in the extracted spin rate for measuring the spin-orbit coupling*
- *Measuring the change in the instantaneous spinning rate $\Delta\Omega$ of a rotor*

We also added the following in the caption of Fig. 6:

The scatter relative to the trend line originates from thermal fluctuations in transient orbits of freely diffusing Brownian rotors (see Supplementary Information).

5. **In a study that emphasizes the importance of HD interactions, it seems the authors ignore a significant body of work dedicated on the emergence of hydrodynamically bound states in systems of rollers, e.g., Martinez-Pedrero et al (Sci. Advances, 2018) Delmotte (Phys. Rev. Fluids, 2019), and spinners (confined to a 2D plane)-in addition to Ref. 24, Goto et al. (Nat. Comm, 2015), Kokot et al (PNAS, 2017). In addition to Ref. 7-9, the ordering in monolayer of rotors has been analyzed by Lushi and Vlahovska (J Non-linear Science,**

2015), where orbital motion in such 2D systems of rotors is predicted (albeit in free space, no wall).

We thank the Referee for drawing our attention to past numerical simulations and experiments investigating hydrodynamically bound states in rollers and spinners. We have added the references to the manuscript, specifically those treating systems of spinners. The work by Lushi and Vlahovska is especially interesting, offering predictions on counter-rotating particles, which the system presented in our work makes experimentally accessible. We specifically emphasize this work (as well as the related work by Kokot *et.al* 2017) by including the following sentence in the conclusion of the manuscript:

Moreover, combining our system of optical rotors with rotors driven by an external magnetic field could enable the experimental study of ensembles of counter-rotating particles, where optical rotors rotate independently from the magnetic rotors. Experimental investigation of an ensemble of counter-rotors would elucidate recent predictions on self-assembly, phase separation, and edge modes, expanding our understanding of far-from-equilibrium states of matter^{12,13,16,17}.

6. In conclusion, while the experiment is neat, the novelty beyond the new experiment is limited. While orbital motion is indeed observed for a first time experimentally, it has been theoretically predicted in systems of plane-confined rotors.

We thank the Referee for finding the experiment to be neat. Our findings indeed show hydrodynamic spin-orbit coupling in a synthetic system for the first time and also offer researchers who pursue emergence in ensembles of coupled rotors a new experimental test bed to revise previous findings where hydrodynamic spin-orbit coupling was inaccessible by construction.

We sincerely thank the Referee for their in-depth comments and suggestions.

Response to Referee 2

1. The manuscript reports a study of the dynamics of a sample of micro-particles immersed in water. The particles are birefringent, and some effort was made with particle synthesis to ensure they were stable in water. The particles can be optically rotated by illuminating them with circularly polarised light - and an unfocussed laser beam is chosen such that the particles are not translationally optically trapped, and so are still able to freely diffuse while rotating. 2D particle tracking is performed allowing the trajectories of the particles, and their hydrodynamic interactions to be analysed. This analysis compares well with analytical theory of low Reynolds number hydrodynamics. In particular, the interaction of pairs of particles is observed, in which the particles become transiently coupled in their translational motion.

I find the work interesting, and as far as I am aware, this is the first such analysis of a complex sample of this nature. The experiments and theoretical analysis appear to be rigorously carried out, however I do think the explanation of the theory could be clearer in places, see more details below. Perhaps I am missing something deeper, however, it does not seem unexpected that a spinning particle should cause nearby particles to orbit. Therefore overall I am unsure of the wider significance of these findings. A few examples are given in the conclusion, however it is unclear where or how the theory developed in this work could be applied to give a deeper understanding of any other situation/type of sample. Therefore, I suggest this wider significance is articulated more clearly for consideration in a broad interest journal such as *nature communications*.

We thank the Referee for finding our work interesting and rigorous. While we agree that pairs of rotating particles are expected to couple hydrodynamically, previous experimental work showed incompatible results: biological micro-rotors^{18,19} and synthetic micro-rotors^{20,21} show different dynamics. This discrepancy manifests itself in large ensembles of rotors, so-called chiral fluids^{1,22} or crystals^{3,20,21}. Moreover, isotropic materials made of rotors are theorized to have unique material properties impossible at equilibrium¹¹, and it is unclear if such materials are experimentally accessible in a non-isotropic system where all particles point in the same direction (such as magnetically stirred particles^{1,22}). Our work focuses on pairs of micro-rotors as a first step in paving the way to study large ensembles of asynchronous rotors in search of novel emergent behavior.

We added the following in the main text to emphasize the broader significance of our findings: The motion of synthetic micro-rotors studied so far is incompatible with the $\mathbf{R} - \theta$ hydrodynamic coupling observed in pairs of biological micro-rotors^{3,18}. Moreover, previous studies with ensembles of synthetic micro-rotors^{20,22} spin by an externally imposed field and can not show spontaneous symmetry breaking as seen in ensembles of biological rotors¹⁹.

2. **line 106: MSD - Mean squared displacement? is not defined. I think should be for a broad audience journal like nat. comms.**

Thank you for the suggestion. We added the following in the main text to define MSD:

Measuring the particles' mean squared displacements (MSDs), $\langle \Delta r^2 \rangle = 4D_t \tau$, we observe a reduction in their diffusion constants D_t compared to the bulk value, $D_t^{\text{bulk}} = k_B T / 3\pi\eta d$ (Fig. 3b).

3. **line 113: The manuscript states ‘..monitoring the particles’ vertical rising speed..’ How is this out of plane motion monitored?**

In the original version of the manuscript, the measurement of the particle's vertical rising speed was explained in the Supplementary Information (see Fig. S3). Following the Referee's question, we added a detailed sub-section in the Supplementary Information:

– *Measuring the average rise velocity v of a particle in the presence of an optical flux*

We also added the following sentences in the main text describing the measurement process:

At higher fluxes where F_{rad} exceeds F_g , vaterite particles begin to steadily rise from the capillary's bottom surface at a constant speed. Vertically shifting the imaging focal plane from the bottom of the capillary to its top surface, we monitor the time it takes for particles to travel $100\mu\text{m}$, corresponding to when a focused image of a particle re-appears (see Supplementary Information for additional experimental details).

4. **line 161: ‘The magnitude of the Fourier of the individual particles,..’ should instead be something like: ‘The magnitude of the Fourier Transform of the intensity transmitted by individual particles as a function of time,..’?**

Corrected – thank you for pointing out this oversight.

5. **line 162: ‘corresponding to four times the typical rotation frequency (0.125 Hz)’ It is not clear to me why this is four times rather than twice the typical rotation frequency?**

For light to reach the detector (camera) through crossed-polarizers, the particle must de-polarize the linearly polarized illumination beam. In a birefringent particle, peak de-polarization happens when the light is polarized at 45 degrees relative to the optical axis of the particle. This happens four times per full revolution^{9,23}. We illustrate this with a cartoon in Fig. 1c. There, the measured light intensity peaks once while the schematic particle has rotated by only a quarter period.

To clarify this in the text, we revised the schematic in Figure 1c and added the direction of the illumination’s electric field (set by the polarizer). The new caption in Figure 1 now reads:

Experimental set-up to drive micro-rotors asynchronously. **a** Optical setup for introducing a broad ($D \approx 440 \mu\text{m}$) circularly polarized beam into a microscope sample. **b** Schematic and polarized microscopy image of birefringent vaterite particles rotating while moving freely in the illuminated region. The transmitted light intensities of two particles (blue and green) are tracked throughout the experiment and are shown in **c** and the inset of **d**. **c** One-half of the particles’ (blue and green) blinking cycle, demonstrating that their optical axes are asynchronous. The incident electric field – whose direction is set by the orientation of the polarizer (P) – is de-polarized whenever the optical axis of the rotating particles is aligned with neither the polarizer nor analyzer (A). **d** Computing the magnitude of the Fourier transform ($\sqrt{\mathcal{F}\mathcal{F}^*}$) of the blinking patterns (inset) of the two particles in **b** shows that the frequencies at which the particles de-polarize the incident L.E.D. light are centered around 0.5 Hz, corresponding to a rotation frequency of 0.125 Hz. However, the magnitude of the sum of transforms, $|\sum_i \mathcal{F}_i|^2$ (solid line), decays, confirming that the particles are out of phase. Scale bar: 5 μm ;

We also added a new subsection in the Supplementary Information entitled:

– *Measuring the frequency and global phase of a rotating particle*

The new subsection provides additional details relating a particle’s rotation frequency to the intensity of light that it transmits over one period. As the 4-fold blinking per revolution is more evident for a birefringent particle that is not perfectly spherical, we attach a video showing this for the Referee’s reference.

6. **line 163:** ‘The asynchronous phases of the light intensities add-up destructively,’ I know what is meant here, however I find this use of the term ‘destructive’ when speaking of purely real positive functions which can’t ‘cancel each other out to zero’ a bit misleading. Consider re-phasing this sentence?

We thank the Referee for pointing out this potential ambiguity. We rephrased the sentence, which now says:

The different phases of the light intensities do not necessarily add up constructively.

7. **Equation 4 - might be helpful to give more detail of how this equation was constructed, maybe in the supplementary.**

In response to the Referee's suggestion, we have included a new section in the Supplementary Information that presents the derivation of Equation 4, entitled:

– *Spinning angular frequency Ω of a birefringent particle*

The section details the relationship between elliptically polarized light's optical torque and a birefringent particle's spinning frequency.

8. **Figure 3b inset: I don't follow why the data points appear to form into two parallel lines. Is this showing particles of different sizes? I think this inset needs more explanation in the caption/main text. Also, what sized particle is this data plotted for?**

We appreciate the Referee for noticing that the caption describing Figure 3 needed additional details. The inset of Fig. 3b serves to show that the translational mean square displacement (MSD) is linear with time (Brownian motion). The diffusion constants presented in the main panel were extracted from the y-intercept of these slopes. Each curve (line) in the inset represents an individual particle of a different size. We added a dashed line in the inset to indicate the expected linear trend. In the main panel, we included grey arrows to indicate the corresponding particle sizes. To describe the process of computing the translational diffusion constant, we have included a new subsection in the Supplementary Information:

– *Measuring the translational diffusion constant of rotors*

Our procedure follows the methods described in Refs. 28 and 29. For completeness, we have added a blue arrow in the main panel of Fig. 3c to indicate the particle size for which the correlation function presented in the inset is measured. The updated caption in the new version of the manuscript now reads:

Inset shows the translational mean-squared displacement (MSDs) for particles with diameters $d = 1.92 \pm 0.11\mu\text{m}$ and $5.92 \pm 0.35\mu\text{m}$. The $\text{MSD} \propto \tau^1$, consistent with particles undergoing Brownian

motion. The translational diffusion constants D_t obtained from these two MSDs are indicated by grey arrows in the main panel. **c** Rotational diffusion perpendicular to the wall, (spinning) $D_{r,\perp}$, measured using depolarization intensity decorrelation, g_{PA} . The blue arrow in the main panel indicates $D_{r,\perp}$ obtained from fitting g_{PA} of a $d = 2.31 \pm 0.14 \mu\text{m}$ particle (inset).

9. **Line 173: The Green's function equation - I suggest all variables (epsilon, r - is at distance in any direction, or distance perpendicular to the rotation axis?) and indices (ijk) are specifically defined here, along with some additional explanation, to help those readers unfamiliar with how the Stokelet and Rotlet descriptions of hydrodynamic interactions operate.**

In the new version of the manuscript, we added a complete description of all variables, and we hope this makes the notation clearer for researchers from different disciplines. On lines 190-193:

We introduce a singularity at position (x_0, y_0, z_0) , acting as a point-torque disturbance (rotlet). The corresponding Green's function, \mathbf{G}_{ij} , satisfying the Stoke's equations is $\mathbf{G}_{ij} = \frac{\epsilon_{ijk} T_k}{r^3}$ ²⁴, where ϵ_{ijk} is the Levi-Cevita symbol whose indices represent components of the rotlet's position in the Cartesian coordinate system, and r is a 3D vector pointing from the rotlet to a point (x, y, z) in space

And on lines 203-205:

Here \hat{x}, \hat{y} are Cartesian unit vectors, and $|R_{\pm}| \equiv \left(x^2 + y^2 + (z \mp \delta)^2\right)^{\frac{1}{2}}$, representing the distance to a point (x, y, z) in space from the source and image charges, respectively. In the far-field limit, $|R_{\pm}|^{-3} \approx \frac{1}{r^3} \left(1 \pm \frac{3\delta^2}{r^2}\right)$.

In the revised version, Fig. 5 also graphically depicts these quantities.

10. **Line 178-187: I suggest clarifying this section. I don't follow how the statement on line 179-180 about the flow field decaying as $1/r^3$ then ties in with the following sentences and eq5 which has a $1/r^4$ scaling. These equations are introduced quickly, without proper definition of the many terms (e.g. $r, \hat{x}, \hat{y}, \hat{\theta}$). Also is \mathbf{R} superscript(+) the same as \mathbf{R} subscript(+)? I assume so but it is a little confusing seeing both. Figure 5 is helpful, and I understand these equations are well used in many body low Reynolds number hydrodynamics (as is the method of images near a boundary), however for readers unfamiliar with this field, I think more explanation is needed.**

We thank the Referee for this comment. We also added a detailed step-by-step derivation in the Supplementary Information, entitled:

– *Flow generated by a rotating sphere near a wall in the Stokes-flow regime.*

In the subsection, we detail the calculation that leads to Eqs. 5 and 8. We also added the following sentence (line 207) in the main text to fully show how the $1/r^4$ scaling arises:

This scaling arises from noting that $-y\hat{x} + x\hat{y} = r\hat{\theta}$.

11. **Line 197: ‘microscopic organism’ ↯ ‘microscopic organisms’.**

We have made this change and thank the Referee for noticing this oversight.

We sincerely thank Referee 2 for their detailed review of our manuscript, helping to clarify our experiments and theory, and making the manuscript more accessible to a broader audience.

Response to Referee 3

1. The manuscript “Hydrodynamic spin-orbit coupling in asynchronous optically driven micro-rotors” presents a synthetic system of active self-rotating particles which have both rotational and translational degrees of freedom.

The manuscript presents a novel biomimetic system of active rotors. Other examples of synthetic active rotors have translational, but not rotational, degrees of freedom because their active rotations closely follow an external field. By contrast, in this manuscript, the particles are able to rotate with different speeds and different phases, much like self-spinning living cells, which interact only hydrodynamically. The manuscript is well written and includes a concrete set of experimental results that convincingly test quantitative predictions for these kinds of active rotors. I recommend publication.

We thank the Referee for finding our manuscript well-written and identifying that the synthetic swimmers presented here resemble living matter.

2. **Before publication, I suggest that the authors consider the following two points: The inset in Fig. 6c was not clear to me. Is it the spinning rate Ω or its change $\Delta\Omega$ which is plotted?**

We thank the Referee for pointing out a potential ambiguity in the caption of Fig 6c. In the new version of the manuscript, the caption reads:

Angular speed ω of two orbiting spheres (diameters $6.7\ \mu\text{m}$ and $5.1\ \mu\text{m}$) at different separations, along with the dependence of the spinning rate Ω on the normalized separation (inset).

3. **What are the theoretical predictions plotted along with the experimental data?**

The theoretical predictions in the inset of Fig. 6c are derived from Faxen’s 1st and 2nd laws (Eq. 6 and 7). Using Eq. 5, the flow-field generated by a single rotor $u(\mathbf{r})$, we compute the dependence of the spinning rate of each rotor, $\Omega_{i,j}$, on the particles’ separation r , given their asymptotic spinning rates at large separation $\Omega_{i,j}^0$. We added the following comment in the caption to clarify this:

Curves show the predicted spinning rates for each particle at different separations – derived from Faxen’s laws (Eqs. 6 and 7 in the main text) – given the particles’ asymptotic spinning rates, $\Omega_{i,j}^0$, measured at large separations

4. **The following recent work may be of interest as an analogous biological system: Odd dynamics of living chiral crystals Tan et al Nature 607, 287 (2022).**

The recent findings by the Fakhri group (now cited in our work) demonstrate a beautiful example of emergence found in asynchronously rotating particles. We aspire that the dialogue between biological and synthetic model systems will enhance our understanding of new states of matter far from equilibrium.

We thank Referee 3 for their kind words and critical comments.

References

- [1] Bililign, E. S. *et al.* Motile dislocations knead odd crystals into whorls. *Nature Physics* **18**, 212–218 (2022). URL <https://www.nature.com/articles/s41567-021-01429-3>.
- [2] Oppenheimer, N., Stein, D. B. & Shelley, M. J. Rotating membrane inclusions crystallize through hydrodynamic and steric interactions. *Physical Review Letters* **123**, 148101 (2019).
- [3] Massana-Cid, H., Meng, F., Matsunaga, D., Golestanian, R. & Tierno, P. Tunable self-healing of magnetically propelling colloidal carpets. *Nature Communications* **10** (2019). URL <https://doi.org/10.1038/s41467-019-10255-4>.
- [4] Aubret, A., Youssef, M., Sacanna, S. & Palacci, J. Targeted assembly and synchronization of self-spinning microgears. *Nature Physics* **14**, 1114–1118 (2018). URL <http://www.nature.com/articles/s41567-018-0227-4>.
- [5] Vogel, R. *et al.* Synthesis and surface modification of birefringent vaterite microspheres. *Langmuir* **25**, 11672–11679 (2009).
- [6] Friese, M. E. J., Nieminen, T. A., Heckenberg, N. R. & Rubinsztein-Dunlop, H. Optical alignment and spinning of laser-trapped microscopic particles. *Nature* **394**, 348–350 (1998).
- [7] Donato, M. G. *et al.* Light-induced rotations of chiral birefringent microparticles in optical tweezers. *Scientific Reports* **6** (2016).
- [8] Leach, J., Mushfique, H., di Leonardo, R., Padgett, M. & Cooper, J. An optically driven pump for microfluidics. *Lab on a Chip* **6**, 735 (2006). URL <http://xlink.rsc.org/?DOI=b601886f>.
- [9] Parkin, S. J. *et al.* Highly birefringent vaterite microspheres: production, characterization and applications for optical micromanipulation. *Optics Express* **17**, 21944 (2009).

- [10] Zhang, S. *et al.* Ultrasensitive rotating photonic probes for complex biological systems. *Optica* **4**, 1103 (2017). URL <https://opg.optica.org/abstract.cfm?URI=optica-4-9-1103>.
- [11] Avron, J. E. Odd viscosity. *Journal of Statistical Physics* **92**, 543–557 (1998).
- [12] Nguyen, N. H., Klotsa, D., Engel, M. & Glotzer, S. C. Emergent collective phenomena in a mixture of hard shapes through active rotation. *Physical Review Letters* **112** (2014).
- [13] Yeo, K., Lushi, E. & Vlahovska, P. M. Collective dynamics in a binary mixture of hydrodynamically coupled microrotors. *Physical Review Letters* **114**, 188301 (2015).
- [14] Oppenheimer, N., Stein, D. B., Ben Zion, M. Y. & Shelley, M. J. Hyperuniformity and phase enrichment in vortex and rotor assemblies. *Nature Communications* **13**, 804 (2022).
- [15] Molloy, J. E. & Padgett, M. J. Lights, action: Optical tweezers. *Contemporary Physics* **43**, 241–258 (2002). URL <https://doi.org/10.1080/00107510110116051>.
- [16] Lushi, E. & Vlahovska, P. M. Periodic and Chaotic Orbits of Plane-Confined Microrotors in Creeping Flows. *Journal of Nonlinear Science* **25**, 1111–1123 (2015). URL <https://doi.org/10.1007/s00332-015-9254-9>.
- [17] Kokot, G. *et al.* Active turbulence in a gas of self-assembled spinners. *Proceedings of the National Academy of Sciences* **114**, 12870–12875 (2017). URL <https://pnas.org/doi/full/10.1073/pnas.1710188114>.
- [18] Drescher, K. *et al.* Dancing volvox: Hydrodynamic bound states of swimming algae. *Physical Review Letters* **102**, 1–4 (2009). 0901.2087.
- [19] Tan, T. H. *et al.* Odd dynamics of living chiral crystals. *Nature* **607**, 287–293 (2022).
- [20] Yan, J., Bae, S. C. & Granick, S. Rotating crystals of magnetic janus colloids. *Soft Matter* **11**, 147–153 (2015).
- [21] Massana-Cid, H., Levis, D., Hernández, R. J. H., Pagonabarraga, I. & Tierno, P. Arrested phase separation in chiral fluids of colloidal spinners. *Physical Review Research* **3** (2021).
- [22] Soni, V. *et al.* The odd free surface flows of a colloidal chiral fluid. *Nature Physics* **15**, 1188–1194 (2019).
- [23] Hecht, E. *Optics* (Pearson Education, Inc, Boston, 2017), 5 ed edn.

- [24] Blake, J. R. & Chwang, A. T. Fundamental singularities of viscous flow. *Journal of Engineering Mathematics* **8**, 23–29 (1974).

REVIEWERS' COMMENTS

Reviewer #1 (Remarks to the Author):

The revised manuscript now clearly shows to novelty of the contribution. I no longer have concerns about the model. I recommend publication.

Reviewer #2 (Remarks to the Author):

The authors have made a good effort to answer all of my questions, and also those of the other reviewers. Furthermore, for me they have articulated more clearly the importance of the work, and so I suggest publication as is.

Reviewer #3 (Remarks to the Author):

The authors have addressed my comments well, and I continue to support publication of this manuscript.